# A Comprehensive Review of Surface Modification Techniques for Enhancing the Biocompatibility of 3D-Printed Titanium Implants

Shuai Long [†] [ID], Jiang Zhu [†], Yiwan Jing, Si He, Lijia Cheng *[ID] and Zheng Shi *

Clinical Medical College & Affiliated Hospital, School of Basic Medical Sciences, Mechanical Engineering College, Chengdu University, Chengdu 610106, China; l1319288735@gmail.com (S.L.); 13378119012@163.com (J.Z.); m18380728911@163.com (Y.J.); hesisi0420@163.com (S.H.)
* Correspondence: chenglijia@cdu.edu.cn (L.C.); drshiz1002@hotmail.com (Z.S.)
† These authors contributed equally to this work.

**Abstract:** The advent of three-dimensional (3D) printing technology has revolutionized the production of customized titanium (Ti) alloy implants. The success rate of implantation and the long-term functionality of these implants depend not only on design and material selection but also on their surface properties. Surface modification techniques play a pivotal role in improving the biocompatibility, osseointegration, and overall performance of 3D-printed Ti alloy implants. Hence, the primary objective of this review is to comprehensively elucidate various strategies employed for surface modification to enhance the performance of 3D-printed Ti alloy implants. This review encompasses both conventional and advanced surface modification techniques, which include physical–mechanical methods, chemical modification methods, bioconvergence modification technology, and the functional composite method. Furthermore, it explores the distinct advantages and limitations associated with each of these methods. In the future, efforts in surface modification will be geared towards achieving precise control over implant surface morphology, enhancing osteogenic capabilities, and augmenting antimicrobial functionality. This will enable the development of surfaces with multifunctional properties and personalized designs. By continuously exploring and developing innovative surface modification techniques, we anticipate that implant performance can be further elevated, paving the way for groundbreaking advancements in the field of biomedical engineering.

**Keywords:** 3D printing; Ti alloy implants; surface modification technology; osseointegration; bone implant

## 1. Introduction

Titanium (Ti) alloy is a highly regarded biomaterial extensively utilized in the field of biomedicine, particularly in orthopedic implants, as illustrated in Figure 1. Ti alloys are used in cranial implants, dental implants, Ti mesh, and artificial joints [1,2]. Ti alloys typically consist of the main component Ti, as well as possible additional elements such as aluminum, vanadium, niobium, and zirconium. Ti as the main component provides the material with lightweight properties and excellent biocompatibility, reducing the risk of rejection reactions. The addition of elements like aluminum and vanadium can enhance the strength and corrosion resistance of the material, increasing its stability when implanted. The inclusion of niobium can improve biocompatibility and reduce tissue reactions, while zirconium alloys exhibit good biocompatibility and aesthetic properties in certain dental applications [1,3–6]. Nonetheless, conventional manufacturing processes for implants, including mechanical machining, injection molding, sintering, and computerized numerical control (CNC) machining [7–9], suffer from certain drawbacks. These limitations include prolonged processing durations, intricate processing procedures, diminished processing precision, and challenges in crafting intricate implant components. These shortcomings are

being effectively addressed through the advancement of 3D printing technology. Within the realm of biomedicine, 3D printing methods such as selective laser melting (SLM), fused deposition modeling (FDM), and direct metal laser sintering (DMLS) have found applications [10–12]. In comparison to traditional machining methods, 3D printing presents a range of advantages, including accelerated production rates, reduced costs, and the capacity to fabricate intricate geometries [13]. At a microscopic level, personalized implants produced through 3D printing demonstrate meticulous control over the porous structure at the implant interface. This diminishes the stress-shielding effect and augments osseointegration [14]. On a macroscopic scale, 3D printing empowers the creation of intricate shapes and structures that are indispensable for tailoring implants to meet specific patient requirements. Certain individuals necessitate exceptionally specialized implant configurations to address unique surgical needs, a feat that proves challenging with conventional manufacturing methods.

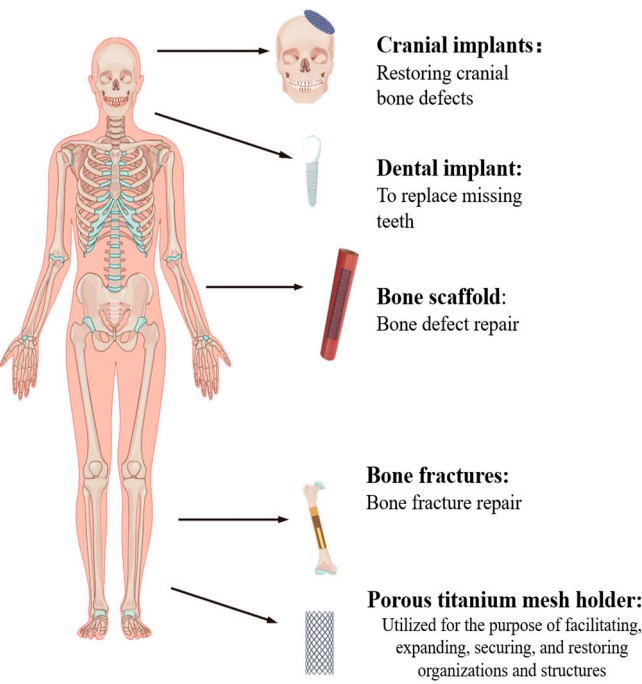

**Figure 1.** Ti alloys used in human body parts.

In summary, the 3D printing of Ti alloy implants delivers swifter and more cost-efficient production while affording the capability to craft intricate and personalized designs at both the microscopic and macroscopic levels. This emerging technology presents substantial advantages over traditional implant-manufacturing techniques.

Although Ti alloys are widely recognized for their excellent biocompatibility, unmodified Ti alloy implant surfaces exhibit biological inertness, making them susceptible to bacterial infections [15]. Additionally, they may suffer from issues like insufficient mechanical stability and poor initial stability performance [16]. To address these challenges, surface modification techniques have emerged as a promising solution.

As depicted in Figure 2, surface modification involves altering the surface properties of a material to meet specific requirements using various methods. Figure 3 categorizes surface modifications for 3D-printed Ti alloy implants into four main types: physical/mechanical, chemical, bio-integration, and multifunctional composite coatings [17].

Physical and mechanical modifications, such as laser treatment, additive manufacturing, and subtractive methods, bring about mechanical alterations to the Ti surface at both microscopic and macroscopic levels [18,19]. Common physical techniques like sandblasting, physical vapor deposition, and laser ablation can enhance tissue integration by modifying surface roughness and topology or by applying physical/mechanical coatings [20].

Chemical modification involves introducing specific chemical reactions or substances to the implant surface. Techniques like anodic oxidation, electrophoretic deposition (EPD), and chemical vapor deposition (CVD) are employed to modify surface chemistry, enhancing bioactivity and interactions with surrounding tissues [21].

Biological coating fusion entails incorporating biologically active molecules or compounds to change the implant's surface composition and structure, imparting specific biological functions [22]. The incorporation of dual-function or multifunction composite coatings, achieved through strategic coating design, enables the achievement of distinct functions in different zones or through gradient modifications. This approach effectively combines antibacterial properties with the promotion of bone growth. The choice of the appropriate surface modification method depends on factors such as the implant's intended purpose, the expected biological response, and the desired mechanical performance.

Additionally, cost, sustainability, and technical feasibility should be considered. Table 1 provides an overview of the characteristics, advantages, and disadvantages of surface modification techniques utilized for 3D-printed implants. Additionally, this article delves into the biological mechanisms underlying these techniques and carefully evaluates their impact on enhancing implant biocompatibility and facilitating bone regeneration. This scholarly piece's primary objective is to provide researchers with a thorough understanding of the surface modification of 3D-printed implants, thereby stimulating the progress of related investigations and fostering the implementation of these techniques in clinical practice. In conclusion, this review provides a comprehensive overview of surface modification techniques for 3D-printed implants, aiming to facilitate further research and the clinical adoption of these emerging technologies.

**Table 1.** Surface modification techniques, advantages, and disadvantages of 3D-printed Ti implants.

| Surface Modification Technology | Advantages | Disadvantages | References |
|---|---|---|---|
| Shot peening/sandblasting | Improves the fatigue and wear resistance of implants. Improves surface hydrophilicity and surface roughness. | Surface has impurities that may cause damage to the surface of the material. | Żebrowski et al., 2019 [23], Bernhardt et al., 2021 [24] |
| LSE | Improved corrosion resistance and mechanical properties, increased surface roughness, and improved biocompatibility and osseointegration. | May lead to surface microcracking and the need to optimize parameters. | Arthur et al., 2023 [25], Simões et al., 2023 [26], Kang et al., 2016 [27] |
| Acid etching | Increasing the surface roughness and improving the surface activity favor the adhesion and growth of osteoblasts and can be used as a pre-treatment. | Time and conditions need to be controlled and over-treatment leads to unstable or damaged surfaces. | Yan et al., 2022 [28], Yu et al., 2020 [29], Ren et al., 2021 [30] |
| Anodization | The formation of an oxide layer to improve osteogenic properties and drug loading to enhance implant biocompatibility. | The high cost of preparation may also affect the mechanical properties of the implant. | Gulati et al. [31], 2017, Maher et al., 2016 [32], Liang et al., 2021 [33], Hunate et al., 2021 [34] |
| EPD | Preparation of the coating on the implant surface results in good surface coverage, more surface material particles, and better coating properties. | Complex coating preparation equipment and processes; the thickness of the coating may not be easily controlled. | Zhao et al., 2022 [35], Teng et al., 2019 [36], Jahanmard et al., 2020 [37], Dian Juliadmi et al., 2020 [38] |

**Table 1.** *Cont.*

| Surface Modification Technology | Advantages | Disadvantages | References |
|---|---|---|---|
| CVD | It promotes osteoblast adhesion and growth by precisely controlling the composition and structure of the coating, providing strong customization, coating uniformity, and durability. | High cost; gas selection and condition control require precision. | Rifai et al., 2018 [39], Youn et al., 2019 [40] |
| MAO | The formation of a dense oxide film and the loading of drugs to improve surface hardness and abrasion resistance, which are conducive to cell adhesion and the growth of bone tissue towards the implant surface and growth. | The bonding strength between the coating and the substrate material may be insufficient, which will weaken its loading capacity. Treatment parameters are difficult to control accurately and the thickness and nature of the oxide layer may be uneven in different areas. | Kozelskaya et al., 2021 [41], Xiu et al., 2016 [42], Sun et al., 2021 [43], Huang et al., 2021 [44], Hu et al., 2020 [45], Tang et al., 2022 [46] |

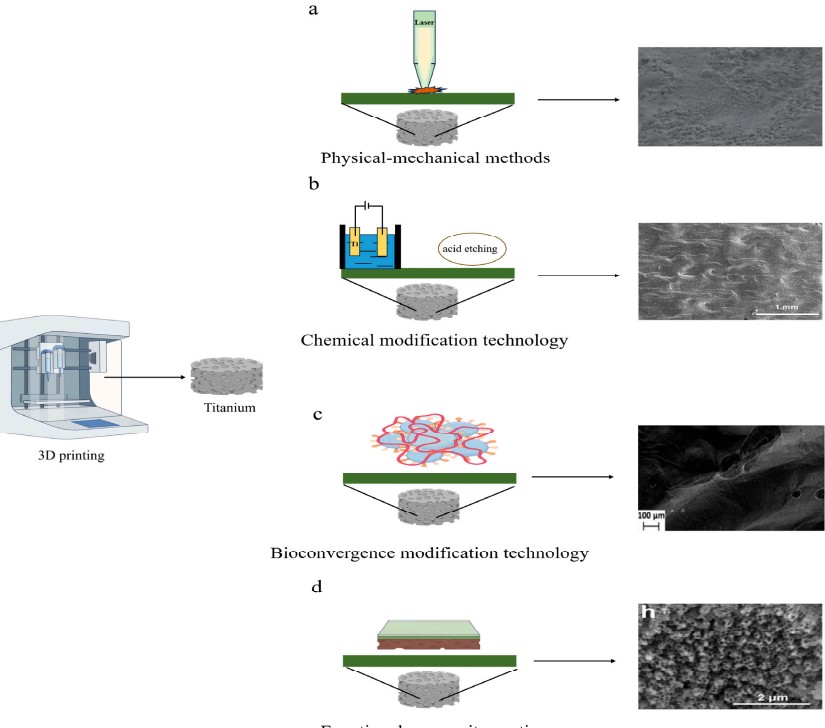

**Figure 2.** Schematic representation of surface modification of physical–mechanical, chemical, bioconvergence, and functional composite coatings. (**a**) Surface microstructure of implants produced using laser surface treatment (Kang et al., 2016) [27]. (**b**) Surface microstructure of implants produced using anodization and acid-etching surface treatments (Ren et al., 2021) [30]. (**c**) Surface microstructure of implants produced using bio-antimicrobial surface treatment (Maver et al., 2021) [47]. (**d**) Surface microstructure of implants produced using composite coating surface treatment (Qin et al., 2018) [48].

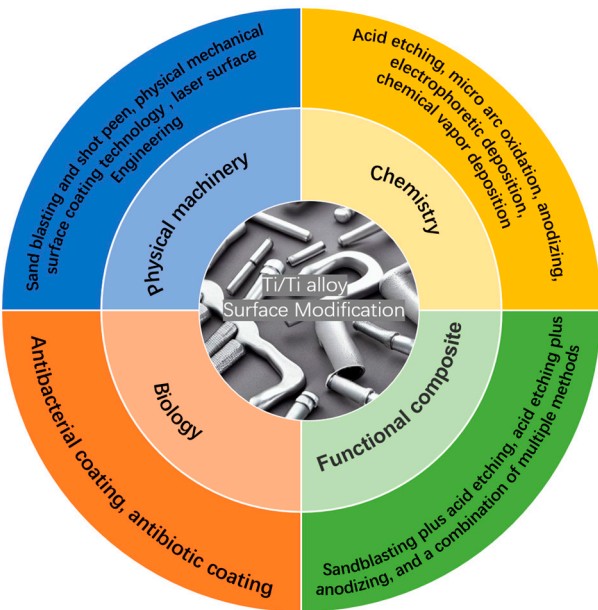

**Figure 3.** Surface modification methods.

## 2. Physical–Mechanical Methods

Physical and mechanical surface treatment methods encompass various techniques that modify the morphology or surface structure of implants. These techniques include sandblasting, peening, surface coating methods, and laser surface engineering (LSE), among others [26,49–51]. The primary objective of these technologies is to improve the biocompatibility, osteoconductivity, wear resistance, fatigue resistance, and long-term stability of implants. It is essential to note that achieving the appropriate surface roughness is a crucial factor in ensuring the longevity of implants [52]. Surface roughness plays a pivotal role in influencing the mechanical stability of implants [53].

### 2.1. Sandblasting and Shot Peen

In the context of 3D-printed Ti alloy implants, a common post-processing technique involves sandblasting. This technology is instrumental in enhancing implant surface properties, imparting a degree of increased surface roughness, and ultimately strengthening the bond between bone and implant [54]. This process contributes to improved osseointegration and enhanced biocompatibility [55]. Frequently utilized materials for sandblasting include steel grit, emery steel shot, and other similar substances [56,57]. Notably, the sandblasting procedure not only fosters bone integration but also generates compressive residual stress on the implant surface, thereby extending its fatigue life [58]. Additionally, it is worth mentioning that sandblasting can be combined with other techniques for even greater benefits. For instance, Chang et al. [59] combined sandblasting with acid etching to enhance the hydrophilicity and roughness of pure Ti and Ti-zirconium alloys, resulting in improved surface properties. Similarly, Bernhardt et al. [24] employed sandblasting to augment the surface roughness of 3D-printed Ti implants.

However, it is important to note that X-ray photoelectron spectroscopy (XPS) analysis has detected elevated levels of exogenous elements such as sodium, magnesium, and silicon on sandblasted surfaces. These elements could potentially be attributed to cross-contamination from the blasting media. Assessments using extracts from the sandblasted surfaces, however, revealed no cytotoxic effects in indirect assays. While sandblasting stands as a straightforward and efficient surface treatment method for enhancing surface roughness without causing cell toxicity, it is crucial to exercise precise control over the sandblasting parameters to prevent any damage to the material. Moreover, consideration must be given to subsequent disinfection and cleaning processes, with a focus on minimizing

any negative impact stemming from the sandblasting. On a different note, another surface treatment technique involves peening, which entails bombarding the implant surface with high-velocity particles, typically abrasives. This process leads to alterations in alloy hardness, improved wear resistance, extended fatigue life, enhanced roughness, and increased corrosion resistance [60]. Shot peening, in particular, proves beneficial for enhancing the fatigue resistance and wear resistance of implants. Żebrowski et al. [23] conducted a study investigating the effects of three different shot peening processes (using CrNi steel shot, crushed nut shells, and ceramic balls) on the surface of 3D-printed Ti alloy. Their findings demonstrated that shot peening strengthens the surface, bolsters the strength parameters, and exhibits low cytotoxicity. Furthermore, the shot peening process increases surface roughness and enhances fatigue resistance [60]. Nevertheless, optimal parameter selection is crucial when considering material design.

In summary, shot peening treatment provides favorable conditions for the clinical application of 3D-printed Ti alloy implants. However, selecting the most suitable surface treatment approach depends on specific implant design and usage requirements. Further investigations involving cellular and animal experiments are essential for a deeper understanding of the impact mechanism and optimization methods of surface treatment on the biological functionality of implants in the future.

### 2.2. Physical Mechanical Surface Coating Technology

Physical mechanical surface coating technology is a widely employed method for modifying the surface of 3D-printed Ti alloy implants. This technique involves the application of specific materials onto the implant surface to enhance its performance, biocompatibility, biomechanical properties, and integration with bone tissue. The coatings used are typically biologically active and facilitate the attachment and growth of bone cells, thereby improving the success rate and durability of the implants. Hydroxyapatite (HA), an inorganic mineral similar to human bone tissue, has been shown to promote the adhesion and growth of bone cells [61]. Fouda et al. [62] coated HA on Ti alloy implants, leading to improved bonding between the implants and surrounding bone tissue and enhanced bone healing. However, different methods of preparing HA coatings can result in bonding strength, crystallinity, and density variations with the substrate. Some methods are prone to delamination, which may cause inflammation at the implant site [63]. On the other hand, HA coatings can achieve excellent integration with personalized porous structure Ti alloy implants [64] and the plasma-spraying technique is generally considered a stable preparation method [65]. Sun et al. [66] demonstrated the successful coating of HA using plasma spraying and electrochemical deposition techniques on 3D-printed porous Ti scaffolds. A comparison revealed that the HA coating achieved smooth and continuous coverage, and the scaffolds with HA coating exhibited exceptional bone repair capabilities. In vitro cell studies confirmed that scaffolds with HA coating have a greater potential to promote the adhesion, proliferation, and osteogenic differentiation of bone marrow mesenchymal stem cells (BMSCs) in the early and late stages. Qin et al. [48] anodized 3D-printed implants to allow the formation of a tunneling nanotube (TNT) on their surface, which was intercalated with HA using an alternative immersion method, and their results showed a significant increase in protein adsorption, cell adhesion, and cell spreading. Expression of the late osteoblast/osteoclast genes gap junction protein alpha 1 (GJA1) and phosphate-regulating endopeptidase x-linked (PHEX) was also enhanced, suggesting cell maturation effects and surface mineralization promotion.

Regarding mechanical properties, Yang et al. [67] found that chitosan composite coatings increased the biomechanical properties but decreased bending resistance. The nontoxic chitosan-based composite coatings facilitated cell proliferation and had good mechanical performance, aiding new bone growth. For early osseointegration, physical coating techniques also provide excellent benefits. Bose et al. [68] incorporated calcium phosphate (CaP) coatings onto 3D-printed porous Ti to improve interfacial bonding between the host bone and the implant surface. The results showed CaP-coated Ti enhanced early

in vivo bone apposition, reducing healing time, with potential applications in orthopedic and dental implantation.

Similarly, Su et al. [69] successfully addressed the bio-inertness of Ti alloy surfaces by constructing strontium calcium phosphate (Sr-CaP) coatings on 3D-printed Ti6Al4V scaffolds. To address the bio-inertness and poor osteointegration of Ti alloys, Zhang et al. [70] explored using CAD combined with 3D printing to reconstruct posterior wall fractures of acetabular fractures, assessing the biomechanical properties of porous Ti alloy scaffolds integrated with steel plates and Ti nitride bio-ceramic coatings. Based on CT scans, NX 10.0 software constructed digital models of the Ti alloy implants with customized open-cell structures. The implants were fabricated via 3D printing, then coated with Ti nitride. The integrated scaffold–plate implants showed excellent matching and biomechanical properties, with good stress distribution and conduction.

### 2.3. Laser Surface Engineering

LSE is a method of material surface processing that utilizes laser technology to selectively melt or alter the surface structure of Ti alloys, which can be applied to modify the surface of 3D-printed implants without changing the properties of the base material itself [71]. When a high-energy laser beam is directed onto a Ti alloy's surface, the laser beam's focused energy generates high temperatures at the irradiation point, causing the rapid or partial melting of the implant surface [72]. LSE enables precise localized processing, allowing for the fine control of the Ti alloy surface at the microscale. Arthur et al. [25] researched using 3D-printed Ti alloys and laser shock treatment to improve the surface properties of Ti alloys, thereby enhancing corrosion resistance and mechanical performance. Simões et al. [26] provided a comprehensive review of the effects of high-power lasers on the performance of Ti alloy implant surfaces, demonstrating that laser treatment can increase the biocompatibility and bone integration ability of Ti alloy surfaces [73]. Lower scanning speeds and higher scan frequencies typically result in increased roughness [74]. Kang et al. [27] found that laser treatment increased surface roughness without compromising fracture toughness while promoting oxygen incorporation to improve Ti wettability. Laser surface modification techniques mainly alter the surface roughness through physical and mechanical means, increasing the contact area to improve the osseointegration and biocompatibility of 3D-printed dental implants. However, LSE can also induce surface microcracks, and improper processing parameters decrease fatigue strength. Thus, an optimization of laser parameters is needed to enhance surface properties comprehensively. In summary, physical and mechanical modification techniques are relatively simple and economical methods to alter the surfaces of 3D-printed Ti alloy implants, more significantly improving mechanical performance, whereas chemical and biological surface modifications focus more on enhancing bioactivity and biocompatibility. The selection and design of a physical coating should align with the specific application, material characteristics, and medical requirements of the implant to ensure successful coating application and clinical outcomes. Sandblasting and peening are suitable for simple implant geometries at lower cost. LSE enables high-precision and complex nanostructures. Overall, compared to chemical and biological modifications, physical and mechanical methods may overly roughen surfaces, negatively impacting biocompatible interfacial bonding strength. Their limited biological activity promotion mainly increases surface roughness to improve mechanics. Future developments may combine physical methods with bioactive surface treatments for synergistic effects.

### 3. Chemical Modification Technology

Chemical methods involve treating the surface of implants with chemicals to induce chemical changes or reactions, thereby altering the surface properties. Standard chemical methods include acid etching, anodic oxidation, microarc oxidation, EPD, and CVD [22,50]. These methods can improve the biocompatibility between the implant and surrounding bone tissue, increasing the implant's success rate and biological performance after implantation.

### 3.1. Acid Etching

Acid etching is commonly used to improve the biocompatibility and osseointegration of Ti alloy implants. Hydrofluoric acid (HF), nitric acid (HNO$_3$), sulfuric acid (H$_2$SO$_4$), or a combination of acids are typically employed to immerse the Ti alloy implants in an acidic solution [75–78], inducing a chemical reaction on the surface that results in the formation of small pits and increased roughness. This enhances surface activity, facilitating the adhesion and growth of bone cells [79–82]. Yan et al. [28] found that acid-etched Ti alloy implants did not significantly affect hydrophilicity but did promote the adhesion and polarization of macrophages with lower levels of reactive oxygen species (ROS). Acid etching is often combined with other surface treatments, such as anodization. Ren et al. [30] combined acid etching with anodization to remove residual powders on the surface, increase surface roughness, and create hierarchical nanostructures with micro pits and grooves. This structural modification improves osteoblast proliferation and osteogenic capacity and enhances new bone formation. Yu et al. [29] combined acid etching and hydrothermal treatment to form a micro/nano-structured surface. The results showed that the microstructure enhanced bone-implant contact, while the nanostructure directly interacted with some cell membrane receptors, providing insights into acid etching as a potential surface modification strategy. Meanwhile, acid etching can also serve as pretreatment. Acid-etched Ti alloy implant surfaces have increased roughness and activity, facilitating osteoblast growth and bone tissue integration improving the success rate and durability of the implant [83]. However, care should be taken to control the treatment time and conditions during acid etching to prevent excessive treatment leading to surface instability or damage.

### 3.2. Anodization

Anodization is widely applied to surface modification of 3D-printed implants due to its simplicity, cost-effectiveness, and versatility [84]. During anodization, the Ti metal implant serves as the anode and is connected to an external power source in an electrolyte solution, creating an electrochemical system. The cathode, typically made of stainless steel or Ti foil, is located at the other end [85]. The process is closely related to the electrolyte, current, and voltage, as well as the composition and surface condition of the metal itself. In addition to using anodization alone, it can also be combined with other methods. Liang et al. [33] used a combination of acid etching and anodization on the surface of 3D-printed implants to induce changes in surface morphology and enhance bioreactivity.

Furthermore, the acid etching and anodization of Ti alloy surfaces have significantly improved osseointegration performance, as shown in Figure 4. During the anodic oxidation of Ti alloys, the typically formed oxide layer is Ti oxide, also known as Ti dioxide. Ti dioxide is a white solid with the chemical formula TiO$_2$ and exists in various crystalline phases, most commonly rutile and anatase [85]. Ti dioxide is widely used in medicine and biology due to its excellent biocompatibility and bioactivity, promoting cell adhesion, proliferation, and bone formation [48]. These advantages make Ti dioxide an ideal surface modification layer, especially suitable for Ti alloy implants to improve compatibility and osseointegration with biological tissues. Gulati et al. [31] formed micrometer-sized spherical particles and vertically aligned TNTs on the surface via anodic oxidation, showing enhanced osteoblast adhesion and the consistent induction of an osteogenic phenotype favorable for the effective osseointegration of Ti substrates. Engineered TNT surfaces can also be used for localized drug delivery in implants. Maher et al. [32] utilized anodic oxidation to create micrometer-sized spherical particles and vertically aligned TNTs on the surface of 3D-printed Ti alloy implants, forming a unique dual morphology. This resulted in the generation of nano reservoirs for drug loading and improving interactions with bone cells. Moreover, TNT exhibited antimicrobial properties [86,87]. Various drugs, including antibiotics, high-concentration anticancer drugs, and antimicrobial agents, were incorporated onto the TNT surface [88–90]. Chunate et al. [34] formed a TNT on the surface of Ti alloy through anodization and loaded 200 ppm of the antibacterial drug vancomycin. The synthesized TNT enhanced the release of vancomycin, with a maximum cumulative release of 34.7%

(69.5 ppm). The Ti alloy implant and TNT displayed excellent wettability. The rough, nanostructured, and nanoporous nature of $TiO_2$ formed on the surface of Ti-6Al-4V holds promise in promoting the biocompatibility and osseointegration of manufactured implants. The composition of the electrolyte, anode material and structure, electrolysis process parameters (voltage, current density, and oxidation time), drug properties, nanotube pore size, and pore structure all influence the drug transport of TNTs. Therefore, it is necessary to consider these factors comprehensively and conduct thorough research and optimization to achieve the desired nanotube array and drug transport effects.

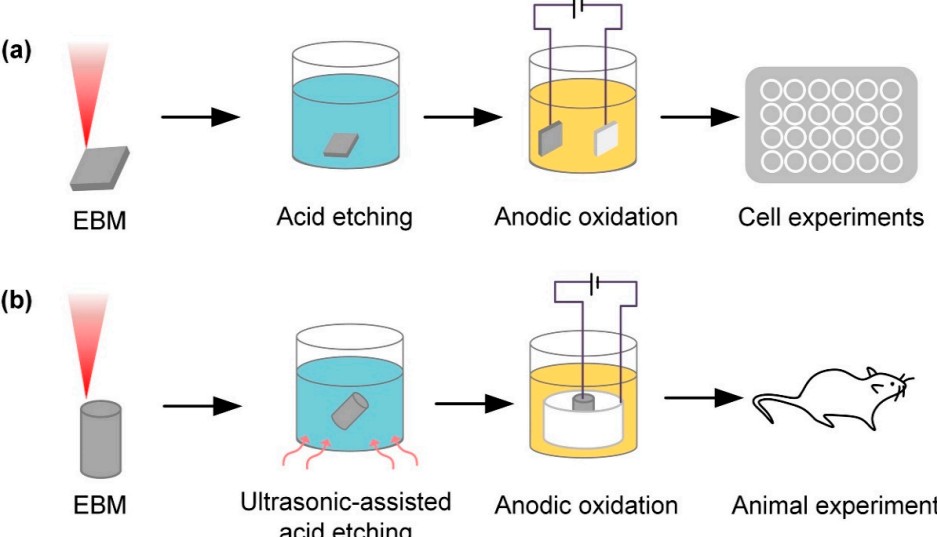

**Figure 4.** Ti alloy implants fabricated using electron beam melting (EBM) technology, with the surface anodized and acid-etched. (**a**) In vitro test after surface acid etching and anodic oxidation treatment. (**b**) In vivo test after ultrasonic acid etching and anodic oxidation treatment (Ren et al., 2021) [30].

### 3.3. Microarc Oxidation

Microarc oxidation (MAO) is an electrochemical method that can form active oxide coatings on metal surfaces. These coatings have excellent biocompatibility and bioactivity [91]. In MAO, the implant serves as the anode and is immersed in an electrolyte solution. An electric current is applied to oxidize the surface. MAO can be applied to more complex implant surfaces, providing a modification strategy for personalized 3D-printed implants. Kozelskaya et al. [41] used MAO to modify the surface of 3D-printed Ti implants with complex internal structures. The results showed that MAO is a valuable method to control the thickness of porous coatings on the internal and external surfaces of 3D Ti implants. The oxidation layer is influenced by voltage, electrolysis time, temperature, and electrolyte composition, which affects implant surface properties [92–98]. MAO treatment has been shown to enhance the biological activity of materials, promoting the growth of bone tissue on the surface of implants. Ni et al. [99] conducted in vitro experiments using MAO-treated 3D-printed porous scaffolds and confirmed that the scaffolds exhibited no cytotoxicity. The biologically active coatings on the surface improved the biocompatibility and bone-bonding ability of the materials. Furthermore, Xiu et al. [42] observed significant improvements in the mechanical properties of implants by applying MAO treatment to the surface of 3D-printed Ti alloy scaffolds. The coated surface demonstrated enhanced osteogenic capability and cell compatibility, providing better mechanical interlocking, increasing the bond strength between bone tissue and implants, and improving wear resistance. Additionally, MAO can form coatings on the surface of implants using specific electrolytes, offering better protection for metallic materials. Sun et al. [43] coated graphene on the surface of porous Ti alloy implants using MAO, significantly increasing surface roughness. In vivo experiments, the graphene-coated group exhibited significantly higher bone ingrowth than the non-coated group, demonstrating excellent bone bonding effects. This provides

good biocompatibility for the implants and further promotes bone bonding effects by increasing the surface roughness of the material. Moreover, standard coatings prepared through MAO include CaP, Sr$^+$, and vancomycin [100–102]; MAO can also form composite coatings, as shown in Figure 5. Furthermore, MAO can be combined with other methods for coatings. For example, Huang et al. [44] studied the surface morphology, chemistry, and cell interactions of coatings prepared via MAO and hydrothermal treatment. In vitro and in vivo results showed that the coated implants enhanced protein adsorption and osteoblast activity, adhesion, and differentiation, promoting early osseointegration compared to implants with MAO alone, improving bioactivity and osseointegration. Hu et al. [45] combined ultrasound with MAO to form coatings on Ti-Cu alloy surfaces, imparting strong long-term antibacterial properties without toxicity. The bonding strength between the MAO layer and metal substrate may need to be improved, compromising load-bearing capability. Processing parameters are difficult to control accurately, with uneven coating thickness and properties in different regions. However, molten salt microarc oxidation is a technique for oxidation in molten salt electrolytes. It can form oxide coatings at lower temperatures, improving the surface roughness of Ti alloy and increasing the content of Ti oxide, thereby enhancing the biocompatibility of implants. Compared to MAO, coatings formed using molten salt microarc oxidation have lower crystallinity, higher density, and better adhesion strength. Additionally, this technique is simple to operate and has low costs. However, the thickness and uniformity of the coatings still need to be optimized. Schwartz et al. [103] utilized plasma electrolytic oxidation (PEO) in aqueous electrolytes and molten salt to obtain biocompatible coatings containing Ti oxide and HA. Compared to samples obtained from aqueous electrolytes, the samples obtained from molten salt exhibited a finer crystal structure morphology.

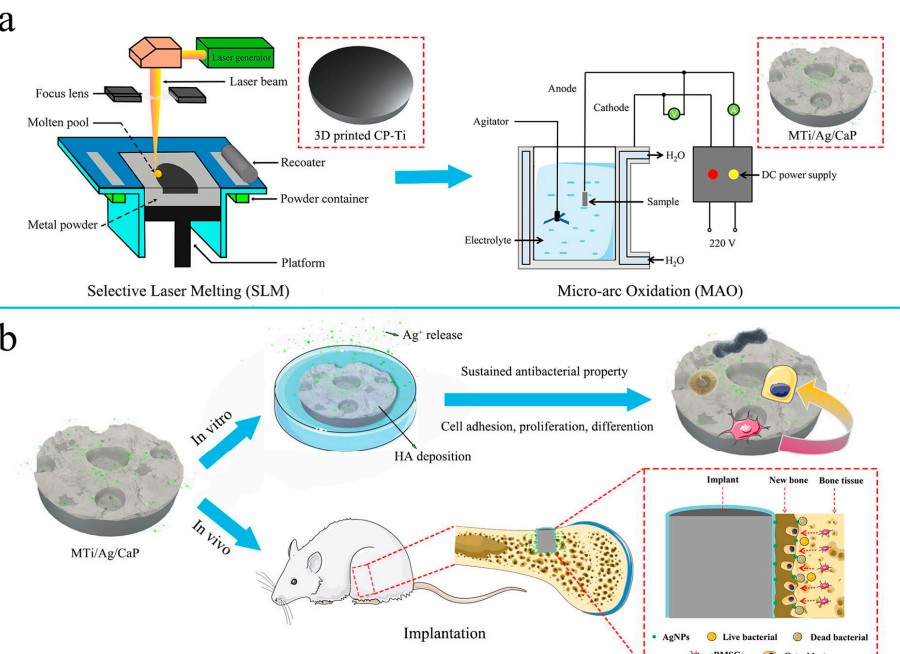

**Figure 5.** Composite-coated implants of apatite decorated with TiO$_2$ nanotubes and Ag nanoparticles. (**a**) A surface-modified Ti implant (MTi/Ag/CaP) with bone integration and antibacterial capabilities fabricated via MAO method, with the outer layer composed of phosphocalcic material decorated with Ag nanoparticles, and the inner layer made of porous TiO$_2$. The MTi/Ag/CaP implant has a bilayered porous structure. (**b**) In vitro experiments showed that MTi/Ag/CaP implant could promote MG-63 cell adhesion, proliferation, and osteogenic differentiation while exhibiting long-term elimination and inhibition of bacterial adhesion and proliferation. In vivo experiments further demonstrated that MTi/Ag/CaP implants generated more mineralized bone tissue than untreated samples (Tang et al., 2022) [46].

### 3.4. Electrophoretic Deposition

EPD is a method of depositing charged particles onto the surface of an electrode. During the process, the charged particles are suspended in a liquid medium, and the electrode is connected to a power source to create an electric field. The charged particles are then deposited onto the surface of the electrode with an opposite charge under the influence of the electric field. This technique has been applied to the surface modification of implants [104]. It can be conducted at relatively low temperatures, making it suitable for heat-sensitive materials. The coating exhibits good uniformity and high controllability, allowing for precise and controlled coating thickness, and it is also suitable for large-scale production [105]. Common EPD coatings include HA, graphene oxide (GO), and Ag [106–110]. Juliadmi et al. [38] deposited natural-source HA coatings onto implant surfaces via EPD, resulting in good surface coverage and more surface material particles, improving coating properties. Furthermore, EPD can also coat more complex shapes. Zhao et al. [111] used EPD to coat 3D-printed Ti alloy meshes with a novel semi-permeable coating for guided alveolar bone regeneration. The coating provided good coverage and revealed antimicrobial effects and biocompatibility. Additionally, Teng et al. [36] found that 3D printing and EPD improved implant surfaces, promoting new bone and blood vessel growth. Over time, the implant slowly released growth factors, better integrating with surrounding bone and improving osteogenesis. EPD can also be used to prepare composite coatings to greatly enhance surface bio-properties [107,112] and bio-organic coatings to improve surface osseointegration [113]. Moreover, EPD has been used to develop special coatings for 3D-printed implants to prevent bacterial infections through effectively killing bacteria and promoting osteoblast growth without cytotoxicity, as shown in Figure 6. This indicates its potential for manufacturing infection-resistant implants in the future.

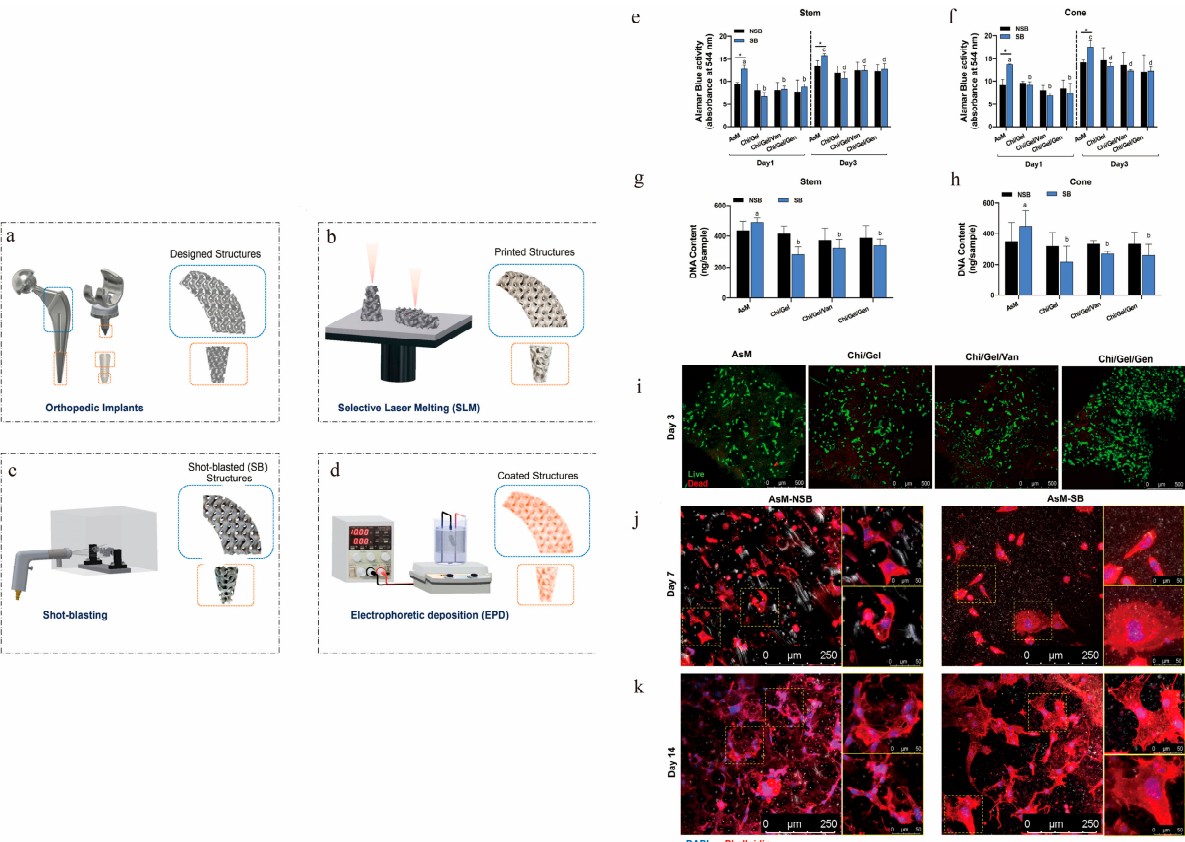

**Figure 6.** Behavior of osteoblast precursor cells with antibiotic-containing coatings on implant surfaces. (**a**,**b**) Ti alloy implants fabricated via SLM. (**c**,**d**) Sandblasting pre-treatment to increase

surface roughness and eliminate surface defects, followed by EPD of antibiotic coatings onto implant surfaces. (**e**,**f**) Alamar blue cell metabolic activity assay after 1 and 3 days of cell culture on the implants. (**g**,**h**) Cell DNA content after 3 days of cell culture on the implants. (**i**) Live/dead staining of stem cell samples after 3 days (live: green; dead: red). (**j**,**k**) Cytoskeletal staining of stem cell samples after 7 and 14 days (DAPI: blue; phalloidin: red). Cytoskeletal staining on days 7 and 14 showed more extensive F-actin tissues on NSB structures; cells on SB structures were more elongated with greater spreading (Jahanmard et al., 2020) [37].) ($p < 0.05$; a with b, c with d, and * shows the significant differences between SB and NSB).

*3.5. Chemical Vapor Deposition*

CVD technology deposits thin films or coatings on a substrate surface through gasphase reactions at high temperatures. In the CVD process, chemical gases are heated and transformed into reactive species, which then react on the surface of the substrate to form solid products [114]. CVD can be employed to enhance the surface characteristics of Ti implants, improving their biocompatibility and functionality. By precisely controlling the composition and structure of the coatings, CVD on the surface of 3D-printed Ti implants can enhance their biocompatibility, promote the adhesion and growth of bone cells, and accelerate the integration of bone with the implant. This technique also offers high customizability, good coating uniformity, and strong durability. Standard CVD coatings encompass HA coatings, which mimic natural bone tissue and enhance the bonding between bone and implants by promoting bone cell growth and formation. CVD can also be utilized to fabricate diamond coatings. Rifai et al. [39] successfully generated diamond coatings on SLM-Ti surfaces through CVD, enhancing cell proliferation and inhibiting bacterial growth. Moreover, composite chemical deposition methods such as metal–organic CVD and microwave plasma-enhanced CVD can enhance the characteristics of coating preparation [115]. CVD surface treatment opens up new avenues, including personalized treatment, improved biocompatibility, and customization options for coatings. As technology advances, CVD is anticipated to play an increasingly significant role in the medical field, offering enhanced implant treatment options for patients. Ti implants have also gained widespread use in bone fracture repair. The CVD technique, specifically initiated CVD, effectively immobilizes proteins and enhances bone cell growth on Ti surfaces. This suggests that CVD may be a valuable bone tissue engineering technology, as illustrated in Figure 7.

In short, chemical surface modification techniques have demonstrated high feasibility in 3D-printed Ti alloy implants. These techniques allow for the selection of suitable methods for surface modification based on implant design and desired performance. For instance, anodization is applicable for enhancing bone integration or loading drugs, EPD is suitable for the precise control of surface coatings, acid etching is effective for surface cleaning and roughness regulation, and CVD is used to achieve multifunctional modification. Compared to physical, mechanical, and biological modification methods, chemical surface modification methods can achieve better biological activity, improve the osseointegration of implants, enhance biocompatibility by altering the surface chemical composition, and cause relatively small damage to the substrate. However, the improvement in mechanical properties is limited, the coating adhesion and stability could be better, some modification methods are complex to operate, and there are high raw material and equipment costs. The control by technical personnel in this field still needs improvement. In the future, optimizing process parameters through digital design can be combined with applying physical and mechanical methods. The overall trend is to achieve precise control over implants' surface morphology, composition, and biological activity.

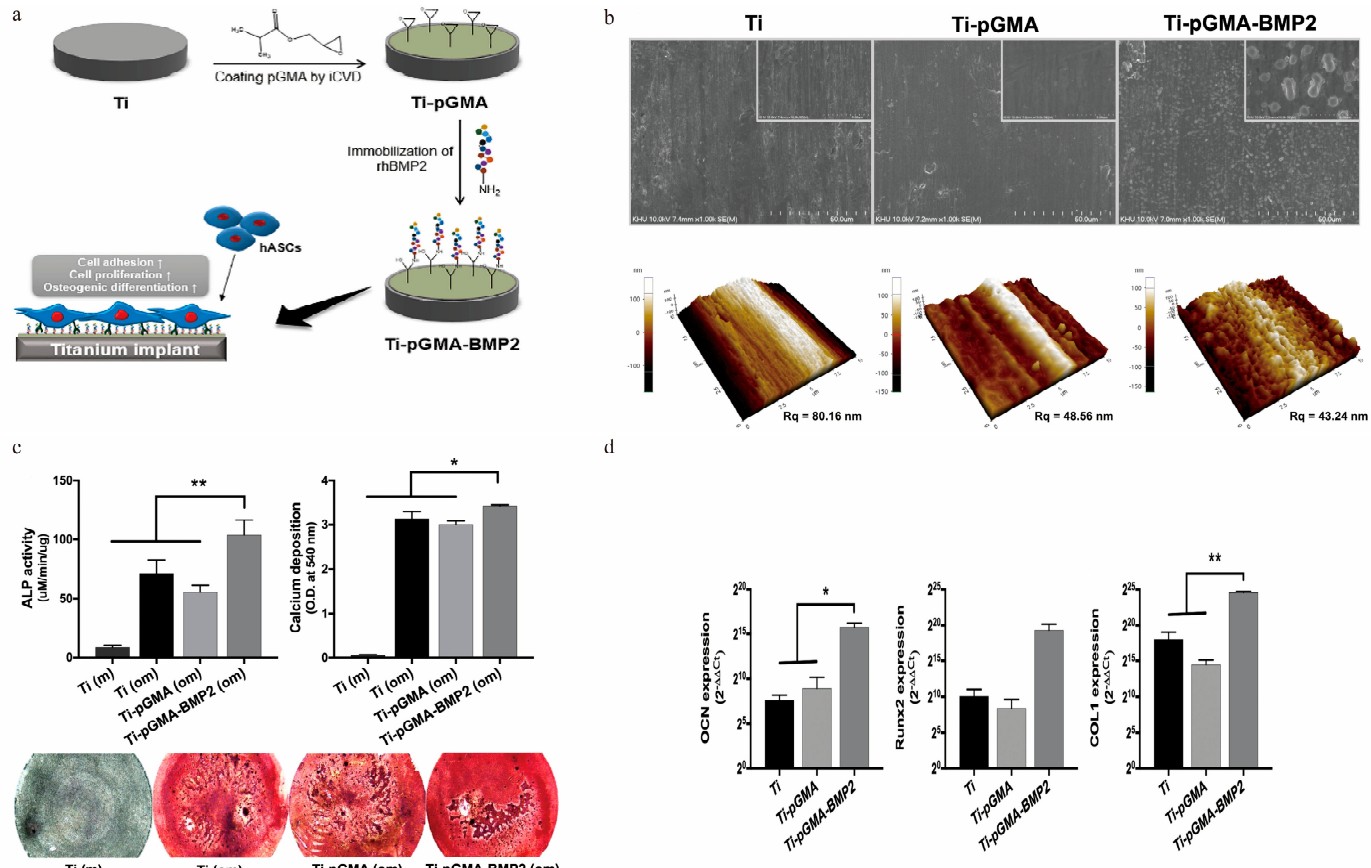

**Figure 7.** Preparation of recombinant human bone morphogenetic protein−2 (BMP−2)-immobilized Ti implants via CVD technique. (**a**) The process of attaching bone growth protein to the surface of Ti implants through CVD. (**b**) Characterization of the Ti surface revealed a uniform 60 nm thick pGMA layer. (**c**) Osteogenic activity of hASCs on Ti, Ti-pGMA, and Ti-pGMA-BMP2 surfaces measured by alkaline phosphatase activity; calcium deposition quantification and Alizarin Red S staining images showed good osteogenic activity in the Ti-pGMA-BMP2 group compared to the other two groups. (**d**) Real-time PCR analysis of osteogenic mRNA gene expression levels in hASCs cultured in osteogenic induction media on the surfaces of Ti, Ti-pGMA, and Ti-pGMA-BMP2, quantifying the expression levels of OCN, Runx2, and COLII as markers of osteogenesis after one day of culture. The mRNA levels of OCN, Runx2, and COLII in the Ti-pGMA-BMP2 group were significantly higher, indicating effective differentiation into osteoblasts (Youn et al., 2019) [40]. (* $p < 0.05$ and ** $p < 0.01$).

## 4. Bioconvergence Modification Technology

Biological modification techniques encompass the utilization of biological processes or components to alter the surfaces of implants. Common biological coatings include antimicrobial, polydopamine, and bioactive organic coatings. These coatings have the capacity to generate tissue-like structures or compositions on the implant surface, thereby fostering cell adhesion, suppressing bacterial proliferation, augmenting bone tissue growth and integration, and enhancing overall implant biocompatibility and bioactivity. Such surface modifications are instrumental in instigating alterations at both the microscopic and macroscopic scales on the implant surface, facilitating a robust connection between the implant and the adjacent bone tissue.

### 4.1. Antimicrobial Coating

Antimicrobial coatings are a type of coating applied to surfaces that is primarily designed to inhibit the growth and reproduction of bacteria and other microorganisms [15]. These coatings typically contain antimicrobial agents or materials, effectively reducing the

survival and spread of bacteria on surfaces, and thus lowering the risk of infection [116]. Commonly used coatings in 3D-printed implants include Ag+ coatings and antibiotic coatings. Ag+ possesses solid antimicrobial activity, interacts with proteins and DNA in bacterial cells, and disrupts the bacterial cell membrane and internal structures, ultimately leading to bacterial death. Silver ion antimicrobial coatings release silver ions upon contact with bacteria, thereby inhibiting bacterial growth [117]. Ag+ can be combined in various ways to form a coating on the surface of implants with good antimicrobial properties and other biological characteristics. Amin et al. [118] loaded $TiO_2$ nanotubes with silver antimicrobial agents through anodization, providing them with additional antimicrobial functionality. The results showed that these biomaterials were highly effective in preventing biofilm formation and reducing the number of planktonic bacteria, especially at intermediate-to-high silver ion concentrations. Despite some cellular toxicity associated with $Ag^+$, its toxicity can be minimized through combination with other methods. Wu et al. [119] modified the surface of 3D-printed implants using plasma PEO and added Ag+ to enhance antimicrobial properties. The coating demonstrated significantly improved antimicrobial performance and bone integration capability without toxicity. Furthermore, Xue et al. [120] developed an efficient, low-toxicity, broad-spectrum antibacterial coating comprising film former polyvinyl alcohol (PVA), polyacrylic acid (PAA), and green-synthesized silver nanoparticles (Ag-NPs) coated onto 3D-printed implants. This coating exhibited excellent antibacterial performance at relatively low concentrations. More importantly, the loading amount and release rate can be controlled by adjusting the Ag-NP content and PVA/PAA ratios, effectively reducing the risk of implant infections. In addition to strong antibacterial properties, implants also need strong osseointegration ability. Surmeneva et al. [121] developed an AgNPs/CaP coating on 3D-printed implants with multiple bio-properties for infected bone defect repair. Ji et al. [22] coated implants with a composite polydopamine and magnesium ion coating, improving the surface wettability and corrosion resistance and reducing the roughness, enhancing the implants' biocompatibility. In summary, silver ion antimicrobial coatings are an up-and-coming antibacterial technology that will continue playing an essential role in medicine, healthcare, food processing, and other areas.

Antibacterial coatings represent a coating technique that applies antibiotics to a surface to inhibit the growth and reproduction of bacteria. These coatings are commonly used on medical devices, implants, and other medical materials to reduce the risk of infection and promote wound or surgical incision healing. Despite its significant advantages in infection prevention, the attachment of antibiotics to the surface of Ti alloy implants requires a carrier, which has drawbacks such as low efficiency and non-biodegradability. Suchý et al. [122] addressed this issue by adding a vancomycin-loaded collagen protein and HA coating through electrospinning on the surface of 3D-printed Ti implants. Their results indicate that this coating can prevent bone destruction associated with Staphylococcus epidermidis infection and enhance bone integration. The coating effectively prevents bacterial infection-related damage to bone structures with minimal systemic load and improves the bone fusion rate.

Furthermore, when antibiotic drugs fail to deliver sufficient amounts to the site of infection (possibly due to systemic administration), they cannot reduce the formation of bacterial biofilms on implants. Maver et al. [47] developed a 3D-printed and electrospun clindamycin-based coating. The results demonstrated that the coating inhibited the proliferation of various bacteria and efficiently transported antibiotics to the infected site, essentially restricting bacterial adhesion and biofilm formation. Guarch et al. [123] also addressed local infections by coating implants with a composite of gentamicin-loaded poly(ε-caprolactone), HA, and halloysite nanotubes, reducing risks of the overuse or misuse of antibiotics leading to resistance. Therefore, antibiotic coatings should be applied prudently with suitable antibiotic types and doses to avoid unnecessary resistance development. Overall, antibiotic coatings are a promising medical technology that can play an essential role in reducing infections and improving treatment efficacy. However, ra-

tional antibiotic use and resistance management should be comprehensively considered during applications.

### 4.2. Other Antibacterial Coatings

Furthermore, there are other antibacterial coatings available. Rifai et al. [124] improved the surface of SLM-Ti scaffolds by applying a nanodiamond (ND) coating, which effectively inhibits bacterial proliferation while also increasing the density of bone cells and fibroblasts. Additionally, a coating combining gallium ions with nitrate was created on a 3D-printed Ti surface to promote bone formation, generate antibacterial properties, and ensure bodily safety [125]. This coating enhances cell adhesion, proliferation, differentiation, and mineralization while preventing the attachment of common bacteria to the surface of 3D-printed porous Ti implants. Incorporating other multifunctional coatings can facilitate cellular differentiation and increase mineralization levels, thereby providing bioactivity to porous Ti implants.

### 4.3. Biologically Active Organic Coatings

Biologically active organic coatings refer to the deposition of protein layers or other organic active coatings on the surface of implants to improve the performance, functionality, or interaction with the surrounding environment of the object. Organic active coatings can be applied in biomedicine, biomaterials, and bioengineering, among others. To address the challenge of poor chemical bonding between implants and bone tissue, organic active coatings can effectively solve the problem. Liu et al. [126] prepared VEGF/BMP-2 core–shell microspheres using coaxial electrostatic spraying technology and loaded the VEGF/BMP-2 core–shell microspheres onto 3D-printed Ti alloy support scaffolds coated with gelatin polymer, thus achieving the sequential release of VEGF and BMP-2 in a composite scaffold system that can effectively promote bone regeneration, providing experimental support and strategies for bone defect repair. Additionally, Guillem et al. [127] functionalized the surface of 3D-printed Ti scaffolds with transgenic elastin-like recombinases (ELRs), and the results showed that the improved surface can enhance bone-bonding ability and regulate cell responses.

On the other hand, the biological characteristics of drug coatings can also be realized. You et al. [128] uniformly coated the surface of a 3D-printed Ti alloy with a drug coating of aspirin/poly (lactic-co-glycolic acid) (ASP/PLGA). In vitro experiments found that the immunomodulatory drug aspirin had a synergistic effect on promoting in vitro osteogenesis and accelerating in vivo bone integration, regulating macrophage polarization, and enhancing osteoblast differentiation and bone integration. In order to optimize the bone growth of Ti implants, Wang et al. [129] used the in-situ sol–gel method to coat CaO on a 3D-printed porous scaffold. Compared to the original bare scaffold, it demonstrated better biocompatibility, cell proliferation promotion, cell adhesion, osteogenic differentiation, mineralization, and bone integration. In addition, the future development trends in organic active coating technology include innovative protein materials, customized coating solutions to meet the needs of different patients, the development of bioactive coatings to promote tissue repair, research on multifunctional coatings to achieve antibacterial and self-healing functions, as well as enhancing coating stability and durability. In addition, chimeric peptides have been utilized to modify the 3D-Ti implant surface to improve the osseointegration of the implant osteo-surface, as shown in Figure 8. Future development trends for bioactive organic coating techniques include innovative protein materials, customized coating strategies to suit different patient needs, developing bioactive coatings to promote tissue repair, studying multifunctional coatings for antibacterial and self-healing capabilities, and enhancing coating stability and durability.

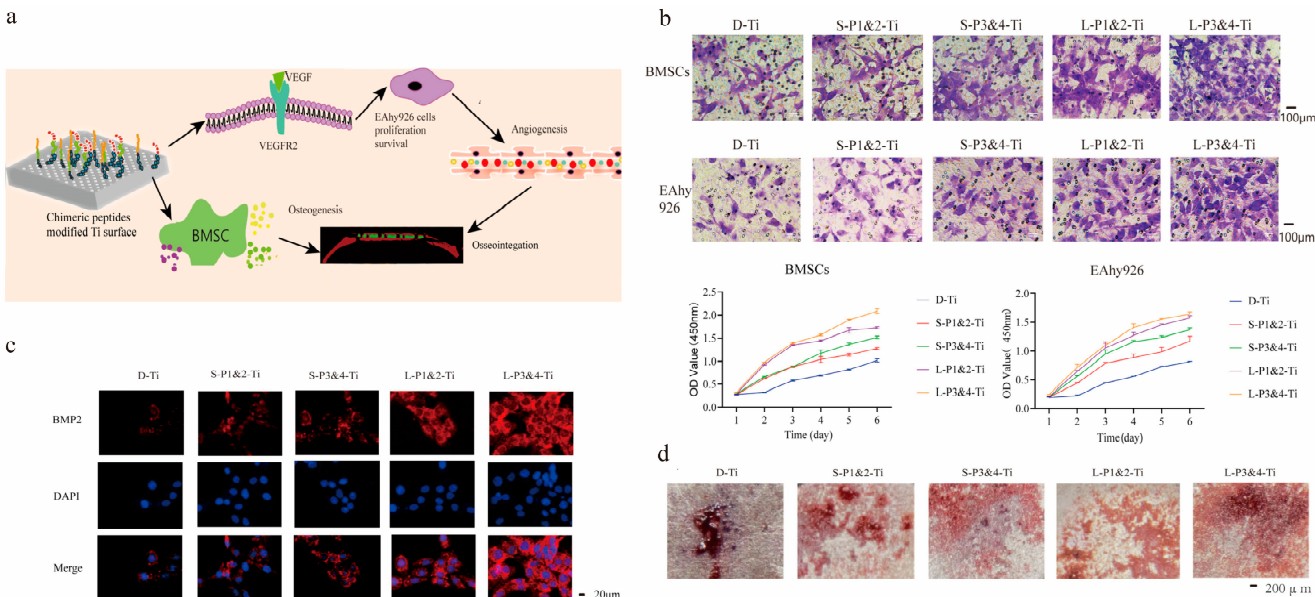

**Figure 8.** Chimeric peptides rapidly modify the surface of personalized 3D-printed Ti implants. (**a**) The schematic diagram illustrates the modification process of the Ti surface using fusion peptides. (**b**) The migration ability of BMSCs and EAhy926 cells cultured on the surface of implants and the proliferation of these cells were significantly enhanced on the fusion peptide-modified Ti surface compared to the D-Ti group. The number of migrated cells in the L-P1&2-Ti and L-P3&4-Ti was higher than that in the S-P1&2-Ti and S-P3&4-Ti, with L-P3&4-Ti group showing more migration of BMSCs and EAhy926 cells compared to L-P1&2-Ti group. (**c**) Immunofluorescence staining revealed an upregulation of vascular endothelial growth factor (VEGF: red; cell nucleus: blue) expression in L-P1&2-Ti and L-P3&4-Ti groups compared to the D-Ti group. (**d**) In vitro studies demonstrated that fusion peptides promote bone integration, with more formation of calcified nodules observed in the L-P3&4-Ti group compared to the L-P1&2-Ti group and significantly more than S-P1&2-Ti and S-P3&4-Ti (Zhao et al., 2021) [35].

### *4.4. Dopamine Coating*

Dopamine coatings are widely utilized functional coatings that enhance implant surfaces' biocompatibility and biological performance. Dopamine, a natural polyamine-based biomimetic adhesive, can form uniform and robust coatings on various materials. In a study conducted by Wang et al. [130], dopamine coatings were applied to 3D-printed Ti implants, resulting in the improved hydrophilicity of the implants and promoting the adhesion, proliferation, and osteogenic differentiation of BMSCs in vitro. Li et al. [131] employed dopamine coatings on porous scaffolds to mitigate stress shielding and facilitate bone growth. Conversely, organic active coatings effectively address implant complications in patients with underlying conditions. Ma et al. [132] discovered that attaching silk fibroin (SF) to the surface of 3D-printed Ti alloy implants reduced the production of ROS and phosphorylation of NF-κBp50 at the bone–implant interface, significantly improving the clinical outcomes of diabetic patients undergoing Ti implantation. As shown in Table 2 each type of coating possesses distinct biological characteristics. These advances will bring more benefits and progress to medical devices, implants, and biomedical applications, opening up new possibilities and breakthroughs in biomaterials and medicine.

**Table 2.** Biological properties of coatings.

| Implant Material | 3D-Printed Method | Coating Materials | Function | References |
|---|---|---|---|---|
| Ti6Al4V | SLM | Ag + coating | Provides strong antibacterial behavior and promotes osteogenesis. | Wu et al., 2021 [119], Amin et al., 2016 [118], Surmeneva et al., 2021 [121] |
| Ti6Al4V | SLM | Antibiotic coating | Inhibits the growth and reproduction of bacteria, reducing the risk of infection. | Maver et al., 2021 [47], Guarch et al., 2022 [123] |
| Ti6Al4V | SLM | HA coating | Improves bone integration ability and osteoinduction; potential for better promotion of bone mesenchymal stem cell adhesion, proliferation, and osteogenic differentiation. | Fouda et al., 2019 [62], Sun et al., 2021 [66], Suchý et al., 2021 [122] |
| Ti6Al4V | SLM | Nano-diamond coating | Inhibits bacterial proliferation and increases the density of bone and fiber cells. | Rifai et al., 2019 [124] |
| Ti6Al4V | SLM | Organic active coating | Effectively beneficial for bone differentiation and osteosynthesis; improves clinical treatment effectiveness for patients with underlying diseases during Ti alloy implantation. | Liu et al., 2022 [126], Guillem et al., 2023 [127], Ma et al., 2021 [132] |
| Ti6Al4V | LENS™ | Cap coating | Improves interface bonding between the bone host tissue and implant surface; reduces healing time by enhancing early bone integration in the body. | Bose et al., 2018 [68] |
| Ti6Al4V | SLM | Polydopamine coating | Forms a uniform and sturdy coating; improves proliferation and osteogenic differentiation; and helps reduce stress shielding and increases bone growth. | Wang et al., 2021 [130], Li et al., 2019 [131] |

These modified coatings have been validated in lab and clinical studies to significantly reduce bacterial adhesion and biofilm formation on implant surfaces, lowering infection risks. They can increase surface amino groups, improving cell adhesion and promoting osseointegration. Specific cell responses can be enhanced through peptide or protein components. However, compared to physical/mechanical and chemical modifications, the stability and mechanical performance are weaker, while physical/mechanical and chemical treatments can greatly improve the mechanical performance. More research and experiments are needed to determine the optimal coating combinations, release rates, and stability. Biological surface modification can improve implants' mechanical performance and bioactivity, representing a future development trend. However, coating processes should be optimized to increase bioactivity stability. Green, degradable, and safe coatings also warrant attention. More advanced biological techniques may be explored for surface functional design.

## 5. Functional Composite Coatings

The composite modification of functional coatings refers to the simultaneous or sequential utilization of two or more methods. Common examples include Ti oxide composite coatings, HA composite coatings, and other composite coatings [133–136]. Different methods are employed to modify the surface for different purposes, giving full play to their respective strengths and combining the merits of various methods to compensate for the defects of a single method. Multi-layer composite structures can be designed to exert the functions of each layer, enhancing antibacterial performance, mechanical properties, and corrosion resistance. The coatings can also improve bioactivity [136]. Improving the interfacial bonding between different layers, such as pre-oxidation treatment followed by coating with a bioactive coating, the oxide layer can provide better bonding to the

underlying layer [137]. However, excessive coating thickness may result in peeling. Some pretreatments, such as alkaline treatment, can effectively enhance the adhesion of subsequent coatings [138]. Heat treatment can ameliorate the stress state between different layers and strengthen the bonding, adopting a progressive design that optimizes each layer step-by-step to obtain a multifunctional composite coating with stable and controllable performance [139]. For instance, Song et al. [140] achieved dual modulation on 3D-printed scaffolds by combining alkaline treatment, heat treatment, and electrochemical deposition of HA coatings to regulate the biological functions of the implant, which markedly improved the stability and bioactivity compared to traditional 3D-printed scaffolds, enhancing osteointegration. For more intricate porous Ti implants, composite methods have exhibited more tremendous advantages. Zhang et al. [141] utilized SLM to fabricate porous Ti alloy scaffolds, followed by sandblasting, acid etching, and atomic layer deposition (ALD) of tantalum oxide films on the surface. Through their functional coating composite method, scanning electron microscope (SEM) morphology and surface roughness tests validated that uniform tantalum oxide films were formed on the inner and outer surfaces of the scaffolds. In vitro results demonstrated that the adhesion, proliferation, and osteogenic differentiation of rat BMSCs were significantly enhanced on the modified Ti alloy scaffolds, ameliorating the cytocompatibility and osseointegration of the porous Ti alloy implants. Additionally, Berger et al. [142] compared sandblasting and acid etching (GB+AE) with GB+AE followed by hot isostatic pressing (HIP) (GB+AE, HIP) by culturing human bone marrow matrix cells (MSCs) for seven days to determine the cell response. The results showed that all exhibited the ability to differentiate MSCs into osteoblasts, with the optimal response of MSCs to the micro/nanostructures produced by the final GB+AE HIP treatment. Apart from composite coatings, surface modification methods can be combined to increase surface bioactivity. Zhang et al. [143] constructed a multi-level micro/submicron/nanostructure by integrating acid etching and anodic oxidation, significantly enhancing surface hydrophilicity, protein adsorption, and biomineralization. Excellent osteogenic performance and increased bone bonding rate were also demonstrated in vivo and in vitro experiments, showing potential for application in personalized bone defect areas.

Similarly, anodic oxidation and acid etching can be applied on the surfaces of 3D porous Ti implants [144]. The composite method of a functional coating is a modified strategy that aims to optimize the comprehensive performance of the surface through scientific design, taking into account the advantages and interactions of various surface treatment technologies. Furthermore, Table 3 summarizes the pros and cons of different types of surface modifications. This is an important reference in current research on the surface modification of 3D-printed Ti alloy implants.

**Table 3.** Advantages and disadvantages of coating preparation methods.

| Type | Advantage | Disadvantage |
|---|---|---|
| Physical–mechanical methods | Physical–mechanical methods are simpler and more cost-effective modifications that can improve the surface roughness and, thus, the osseointegration of the implant, improving the mechanical properties of the surface more significantly. | Physical–mechanical methods may induce poor bioadaptation and interfacial adhesion, have a low capacity to enhance bioactivity, and have limited bioactivity promotion ability. |
| Chemical modification technology | Chemical surface modification methods can achieve better bioactivity results, improve the osseointegration of implants, improve bioadaptability by changing the chemical components of the surface, and be less damaging to the substrate. | Chemical surface modification methods to improve the mechanical properties are limited, coating adhesion and stability are poor, some modification methods are complicated to operate, the cost of raw materials and equipment is high, and the control of the technicians in this field still needs to be improved. |

**Table 3.** *Cont.*

| Type | Advantage | Disadvantage |
|---|---|---|
| Bioconvergence Modification Technology | Promotes cell adhesion, inhibits bacterial colonization, enhances bone tissue growth and integration, and improves the biocompatibility and bioactivity of the implant, making changes to the implant surface at the microscopic and macroscopic levels in order to promote a strong bond between the implant and the surrounding bone tissue. | Biofusion modification technology is less stable and mechanically robust than physical–mechanical and chemical modification methods, and the technology is more cumbersome to operate. |
| Functional Coating Lamination | Composite methods for the different purposes of surface modification have their respective advantages, and the advantages of a variety of methods make up for the shortcomings of a single method. A multi-layer structure can be designed to give full play to the functions of each layer to improve antibacterial properties, mechanical properties, and corrosion resistance, and a coating can improve biological activity. | Composite functional coatings and Ti alloy substrates are poor and easy to peel off; the coating performance is not uniform; the processing technology is complex; and the long-term compatibility of composite coatings with the human physiological environment and other issues remain to be confirmed by further research. |

As shown in Table 4, the physical–mechanical method can produce rough coatings, improving mechanical fixation but with limited biological activity. The chemical method can achieve uniform coatings with improved biocompatibility, but lower adhesion and stability. The biological method directly implants bioactive molecules such as growth factors to achieve specific biological functions, but the mechanical performance of the coatings is weaker as they are more brittle. The future development direction is to design multi-layer composite coatings that leverage the advantages of each layer while considering biological activity, mechanical performance, and antimicrobial properties.

**Table 4.** Structural properties of coatings.

| Coating | XRD | XPS | SEM | Corrosion Resistance | Bioactivity | Disadvantage |
|---|---|---|---|---|---|---|
| Ag+ coating | Diffraction peaks from silver crystals in coatings. | Appearance of silver elemental peaks. | Usually distributed as tiny particles on the surface; white or gray in color. | Achieves some improvement. | Powerful antibacterial activity. | Some cytotoxicity. |
| Antibiotic coating | May show a flat background rather than sharp diffraction peaks. | Characteristic peaks of the antibiotic elements involved, such as sulphur, oxygen, and nitrogen, can be detected. | May be unevenly distributed with areas of aggregation; color may be close to untreated implant surface. | No significant change. | Prevents infections and inhibits the growth of a wide range of bacteria. | May develop bacterial resistance. |
| HA coating | Characteristic diffraction peaks of HA can be detected. | The characteristic peaks of the elements phosphorus and calcium can be seen. | Forms a homogeneous film, which may appear grayish white in color. | Poor corrosion resistance. | Ability to promote bone cell adhesion and growth. | Poor mechanical properties; easily falls off. |

**Table 4.** *Cont.*

| Coating | XRD | XPS | SEM | Corrosion Resistance | Bioactivity | Disadvantage |
|---|---|---|---|---|---|---|
| ND coating | Diffraction peaks of visible diamonds. | Characteristic peaks of visible carbon. | May be highly dispersed or may form agglomerates; bright, grayish, or blackish in color. | Typically high corrosion resistance. | The promotion of osteoblast growth and osseointegration. | Complex process with high cost. |
| Organic active coating | Organic coatings usually do not have a crystal structure and have no visible crystal diffraction peaks. | Characteristic peaks of elements in proteins such as carbon, oxygen, nitrogen, and sulphur can be detected. | Uniform distribution of organic protein coatings; color close to untreated implant surface. | Poor corrosion resistance. | Positive effects on cell adhesion, biomolecular interactions, etc. | Low corrosion resistance. |
| Cap coating | Clear characteristic peaks. | The characteristic peaks of elemental Ca and P can be seen. | Microstructure showing the surface morphology and particle distribution of Cap coatings is usually varying shades of gray. | Better corrosion resistance. | Potential promotion of bone tissue growth and osseointegration. | Susceptible to mechanical abrasion or flaking. |
| Dopamine coating | Amorphous; no obvious diffraction peaks. | A characteristic peak of a high concentration of nitrogen can be seen. | Highly uniform coverage; color may be close to the implant base: slightly darker or shiny. | Poor corrosion resistance. | Promotes improved osseointegration, proliferation, and osteogenic differentiation. | Relatively poor corrosion resistance. |

## 6. Clinical Significance

The surface modification technology of 3D-printed Ti alloy implants has significant importance in clinical applications. Firstly, surface modification can greatly improve the biocompatibility between the implant and surrounding bone tissue, promoting bone tissue growth on the implant surface and enhancing the bond between the implant and bone tissue, thus improving the long-term stability of the implanted device [145]. Secondly, surface modification can enhance the implant's antibacterial properties and reduce the risk of infection, which is crucial in reducing postoperative complications [146]. This can improve the patient's postoperative recovery process, alleviate pain, and increase the success rate of the surgery. In addition, by appropriately treating the surface roughness, the contact area between the implant surface and bone tissue can be effectively increased, promoting the speed of bone tissue ingrowth and contributing to faster healing and recovery [145]. Finally, surface modification technology can also impart drug release functionality to the surface of implants, allowing for localized delivery to surrounding tissues and improving treatment outcomes. This technique can be used for targeted drug release, helping to alleviate patient pain, promote healing, and reduce complications. Overall, surface modification is one of the key technologies ensuring the successful implantation and functionality of 3D-printed Ti alloy implants, with significant clinical implications for enhancing both short-term and long-term postoperative prognosis. The application of this technology provides patients with more personalized, safe, and effective implant options, thereby improving their quality of life and speed of recovery.

**7. Future Directions and Challenges**

The future development trends in the surface modification of 3D-printed Ti alloy implants will encompass physical, mechanical, chemical, and bioconvergence modifications and functional composite coatings, as shown in Figure 9. Furthermore, the future development of physical–mechanical modification technology, with the promotion of interdisciplinary integration, will enable achieving the controllability and designability of the surface morphology and performance of implants through precise control, intelligent computer design, the introduction of new technologies, and personalized manufacturing. Concurrently, chemical modification methods will utilize more bioactive materials and nanotechnology to promote cell adhesion and osteoblast proliferation, enabling more effective tissue regeneration and repair. Furthermore, bioconvergence modification technology will incorporate cell engineering and genetic engineering techniques to directly implant cell adhesion factors or growth factors, expediting the integration of the implant with the host bone. A potential method for the surface modification of orthopedic implants with promising prospects is functional composite surface modification. In the future, personalized treatment will become mainstream, as each patient's orthopedic needs are unique. Some individuals may require improved bone cell growth, while others may need better wear resistance or infection resistance. Composite methods can customize the implant surface based on the specific needs of the patient, thereby achieving personalized treatment. This will help improve surgical success rates and patient satisfaction. Additionally, a single surface modification method may not be able to provide all the required performance characteristics. For example, a single anodization may not simultaneously achieve antibacterial and adhesive properties. However, coatings with an antibacterial layer after anodization can effectively enhance the implant's antibacterial properties and cell adhesion. On the other hand, drug delivery and infection prevention on the implant's surface are crucial. Composite methods can combine drug delivery systems with bio-coatings to prevent infection or accelerate healing. This is essential for the management of long-term implants. Composite surface modification encourages researchers to constantly seek new combinations and techniques to meet evolving patient needs. With the continuous progress of science and technology, new surface modification methods and materials will continue to emerge. Composite methods provide flexibility and feasibility for integrating these new technologies in the future. In summary, composite surface modification has the advantages of adaptability, great potential for personalized treatment, and overcoming the limitations of single methods. Therefore, it is expected to become the most promising surface modification method in the field of orthopedic implants in the future. This will help improve the performance of implants, reduce complications, and improve patients' quality of life. However, these advancements necessitate rigorous scientific validation and clinical practice to ensure their safety and efficacy.

Firstly, long-term stability is a critical issue, as modified layers may lose functionality due to wear, delamination, or material aging, compromising the long-term performance of the implant. Secondly, biocompatibility risks also warrant attention as new materials or bioactive molecules may elicit allergic reactions or tissue rejection responses, impeding the integration of the implant with surrounding tissues. The lack of unified standardized assessment methods makes comparing different study outcomes and determining optimal modification techniques difficult, posing certain barriers to promotion and application. Concurrently, the cost and complexity of modification techniques also limit their widespread adoption as some advanced methods may require expensive equipment and intricate processes, which may not be feasible in resource-scarce regions or healthcare systems.

Long-term clinical validation is crucial to ensuring the safety and efficacy of modified implants. However, this necessitates extensive clinical trials and longitudinal follow-up studies that are time-consuming and costly. Another challenge is accounting for individual variations. The physiological characteristics and needs differ for each patient, necessitating customized surface modification schemes, which adds to the complexity of manufacturing and implementation. To overcome these challenges, interdisciplinary collaboration is

imperative, integrating expertise from diverse fields such as biology, materials science, engineering, and medicine. This can catalyze the establishment of standardized assessment methods to ensure consistency and comparability of modification techniques. Moreover, long-term clinical research should be reinforced to thoroughly understand the performance and effects of modified implants in different patients. Concurrently, efforts should be made to reduce the cost and complexity of modification techniques to make them more accessible and sustainable. By comprehensively addressing these challenges, advancements in surface modification techniques for 3D-printed Ti alloy implants can be stimulated, providing patients with safer and more effective medical options.

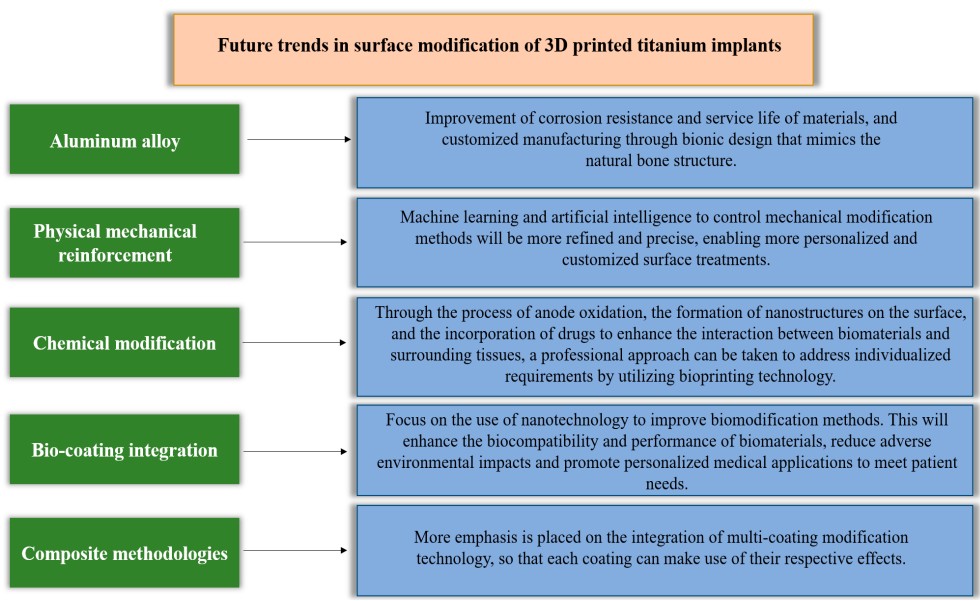

**Figure 9.** Future trends in surface modification.

## 8. Summary

This paper explores various surface modification methods for 3D-printed Ti alloy implants, covering detailed discussions on the principles, processes, functions, and application fields of modification. The effects of applying different coating materials and composite modification methods on osteogenesis, antibacterial properties, bioactivity, and cell proliferation are elaborated. These findings provide valuable references for optimizing the design and performance of orthopedic implants. Based on the research results, future studies will focus on determining the optimal parameters and most suitable coating materials for surface modification techniques to further enhance the fusion and biocompatibility of implants with the host bone. In addition, the establishment of standardized evaluation methods will also facilitate comparison between different study results, promoting development and applications in this field. In summary, this paper offers an in-depth understanding of surface modification for 3D-printed Ti alloy implants and provides beneficial guidance for improving and optimizing implant design in the future.

**Funding:** This work was supported by the Natural Science Foundation of Sichuan Province, China (2022NSFSC1510), the Medical Scientific Research Project of Chengdu City, China (2021043), the Sichuan Provincial Science and Technology Foundation (22NZZH0031), and the higher education talent training quality and teaching reform project of the education department of Sichuan Province, China (JG2021-1102).

**Institutional Review Board Statement:** Not applicable.

**Data Availability Statement:** Data is unavailable due to privacy.

**Conflicts of Interest:** The authors declare no conflict of interest.

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
