# Peer review of "A Comprehensive Review of Surface Modification Techniques for Enhancing the Biocompatibility of 3D-Printed Titanium Implants"

_coatings, doi:10.3390/coatings13111917_

Round 1

Reviewer 1 Report

Comments and Suggestions for Authors

Dear authors,

Your work is interesting, but some adjustments need to be made.

1. To increase readers' interest, adding the method of micro-arc oxidation in molten salts to the work is necessary: https://doi.org/10.3390/ma16134624.

2. In my opinion, it is necessary to introduce a paragraph comparing surface improvement methods, that is, make a detailed comparison in terms of the structural characteristics of the coating (XPS, XRD, SEM, corrosion, adhesion) and their bioactivity.

3. Based on the paragraph comparison of surface improvement methods, the conclusions should identify the most promising surface modification method and briefly describe its possible application in the technology.

Author Response

We feel great thanks for your professional review work on our article. The following are our replies to the comments, and the corresponding changes have been highlighted using the "Track Changes" function in Microsoft Word.

Comments 1: To increase readers' interest, adding the method of micro-arc oxidation in molten salts to the work is necessary: https://doi.org/10.3390/ma16134624.

Response 1: We thank the reviewer for the meaningful suggestion. We agree with this comment. We have taken steps to make the changes you suggested. we describing the method of micro-arc oxidation in molten salts and citing relevant literature. Where in the revised manuscript this change can be found – page 12, section3.3, and line 366-378.

Comments 2: In my opinion, it is necessary to introduce a paragraph comparing surface improvement methods, that is, make a detailed comparison in terms of the structural characteristics of the coating (XPS, XRD, SEM, corrosion, adhesion) and their bioactivity.

Response 2: We thank the reviewer for the suggestion. Your suggestions will help to improve the quality of our articles and make them more comprehensive and useful. We would be happy to take on board your suggestions, especially when comparing different surface improvement methods with more detailed comparisons of structural properties and bioactivity of coatings. We have added a new paragraph and table the article dedicated to comparing various surface modification methods, including XPS, XRD, SEM, corrosion and adhesion properties, as well as their biological activities. This will deepen the reader's understanding and help to better assess the suitability of these methods for different applications. Where in the revised manuscript this change can be found – page 22-24, section 5, and line 692-701.

Comments 3: Based on the paragraph comparison of surface improvement methods, the conclusions should identify the most promising surface modification method and briefly describe its possible application in the technology.

Response 3: We thank the reviewer for the suggestion. we add the most promising surface modification methods and briefly describe their possible technical application areas. This will help to provide readers with clearer conclusions and inspiration for a better understanding of the practical value and potential applications of the research. Where in the revised manuscript this change can be found – page 25, section 7, and line 738-761.

Reviewer 2 Report

Comments and Suggestions for Authors

Write an additional paragraph on clinical implications

Author Response

We feel great thanks for your professional review work on our article. The following are our replies to the comments, and the corresponding changes have been highlighted using the "Track Changes" function in Microsoft Word.

Comments 1: Write an additional paragraph on clinical implications

Response 1: We thank the reviewer for the meaningful suggestion. We will follow your suggestions to ensure the completeness and usefulness of the article. We add that implant surface modifications affect clinical practice and may mention how these surface modifications can be applied in a clinical setting to enhance treatment outcomes or improve patient quality of life. Where in the revised manuscript this change can be found – page 24, section 6, and line 702-722.

Reviewer 3 Report

Comments and Suggestions for Authors

The manuscript entitled “To improve the biocompatibility of 3D printed titanium implants by surface modification: a review” mainly focuses on the review of surface modification techniques for Ti-based alloy three-dimensional (3D) printing technology.

 This manuscript is well written, with a formulated key problem of the corresponding research object. Nevertheless, I have only a few comments on several aspects.

 The term "CNC machining" is not defined.

The quality of the Fig. 3 is very poor. The notes are not visible. A different layout is required for the detailed review of the presented images.

A section about the composition of titanium alloys used for 3D printing is highly required.

Author Response

We feel great thanks for your professional review work on our article. The following are our replies to the comments, and the corresponding changes have been highlighted using the "Track Changes" function in Microsoft Word.

Comments 1: The term "CNC machining" is not defined.

Response 1: We thank the reviewers for their meaningful suggestions. We note that you mentioned that the term "CNC machining" was not clearly stated in the article, and we have added the full name "CNC" to ensure that readers understand that this refers to computer numerical control machining. Where in the revised manuscript this change can be found – page 1, section 1, and line 44.

Comments 2: The quality of the Fig. 3 is very poor. The notes are not visible. A different layout is required for the detailed review of the presented images.`

Response 2: We thank the reviewers for their meaningful suggestions. We have improved the quality of Fig. 3 by increasing its clarity and ensuring that the annotations are more legible. Furthermore, to improve the flow of the article, we have repositioned the sequence from Fig. 3 to Fig. 2. Where in the revised manuscript this change can be found – page 5, section 1, and line 122.

Comments 3: A section about the composition of titanium alloys used for 3D printing is highly required.

Response 3: We thank the reviewers for their meaningful suggestions. We have added a paragraph that describes in detail the composition of titanium alloys used for 3D printing. This includes the content of various elements in the alloy, the strength and corrosion resistance of the alloy, and how these properties relate to 3D printing technology. Where in the revised manuscript this change can be found – page 1, section 1, and line 33-39.

Reviewer 4 Report

Comments and Suggestions for Authors

In the manuscript titled “To improve the biocompatibility of 3D printed titanium implants by surface modification: a review” by Shuai Long and colleagues, the authors comprehensively discussed the range of methods employed in surface modification techniques to enhance the performance of 3D-printed titanium alloy implants. These methods encompass physical-mechanical methods, chemical modification methods, bioconvergence modification technology, and functional composite methods. The review provides in-depth discussions on the underlying principles, procedures, functionalities, and application domains of these modifications, offering insights into their distinct advantages and limitations. The review is thoughtfully structured, with well-organized subtitles and sentences. Overall, this review contributes to a profound understanding of surface modification for 3D-printed titanium alloy implants and serves as valuable guidance for future enhancements and optimizations in implant design, with potential transformative implications in the field of biomedical engineering. However, I suggest the authors address the comments before accepting the manuscript.

Minor comments

-        In Figure I, within the "Physical Machinery" section, the authors referred to "physical coating technology coating." I believe the authors intended to specify "Physical mechanical surface coating technology" instead. Additionally, I observed an inconsistency in the capitalization of method names, where "Physical Machinery" is in the title case, while others, such as "Functional composite," are not. To enhance clarity and maintain consistency, please review, and ensure uniform formatting for all method names.

-        Line 117, the abbreviation "Titanium alloy (Ti alloy)" can be somewhat confusing throughout the manuscript. Would it be more precise to use the abbreviation "Titanium (Ti) alloy" instead?

-        To enhance readability, consider relocating Table 1 to the end of the section discussing the CVD method or right before the Bioconvergence Modification Technology. Alternatively, if you opt to keep it in its current position, please provide expansions for LSE, EPD, CVD, and MAO within the table itself and include horizontal lines beneath each method to minimize reading confusion. Furthermore, it would be beneficial to include the advantages and disadvantages of Bioconvergence modification technology within the table for a more comprehensive overview.  

-        There are abbreviation and expansion-related issues existing in the manuscript. For instance, in section 3.5, Chemical Vapor Deposition is abbreviated as CVD multiple times within the same paragraph. Similarly, there are instances where abbreviations like TNT are used without prior expansion at their initial mention. I recommend a thorough review of the manuscript to ensure that all abbreviations are adequately expanded upon their first use.

-        I suggest moving Table 3 right before the Future direction and challenges.

Author Response

We feel great thanks for your professional review work on our article. The following are our replies to the comments, and the corresponding changes have been highlighted using the "Track Changes" function in Microsoft Word.

Comments 1: In Figure I, within the "Physical Machinery" section, the authors referred to "physical coating technology coating." I believe the authors intended to specify "Physical mechanical surface coating technology" instead. Additionally, I observed an inconsistency in the capitalization of method names, where "Physical Machinery" is in the title case, while others, such as "Functional composite," are not. To enhance clarity and maintain consistency, please review, and ensure uniform formatting for all method names.

Response 1: We thank the reviewer for the meaningful suggestion. With regard to terminological precision, you suggested changing "physical coating technology coating" in the "Physical Machinery" section to "physical mechanical surface coating technology". "Physical mechanical surface coating technology" in the "Physical Machinery" section. We strongly agree with this suggestion as it more accurately reflects the technology discussed in our article. We will make the appropriate changes in the revised manuscript to ensure accuracy and clarity. Second, regarding formatting consistency between titles and method names, you pointed out the use of capital letters in the titles, which are not used in the method names. We recognize this inconsistency and will make the necessary changes in the revised draft to ensure that all method names are presented in a consistent manner, which will help improve the overall consistency and readability of the article. In addition, in order to improve the flow of the article, we have moved the order of Figure 1 to Figure 3. Where in the revised manuscript this change can be found – page 5, section 1, and line 119-120.

Comments 2: Line 117, the abbreviation "Titanium alloy (Ti alloy)" can be somewhat confusing throughout the manuscript. Would it be more precise to use the abbreviation "Titanium (Ti) alloy" instead?

Response 2: We thank the reviewers for their meaningful suggestions. In response to your comments, we have revised the article by changing the abbreviation "titanium alloy (Ti alloy)" to "titanium (Ti) alloy" in the first section and by verifying all abbreviations in the article to improve clarity and accuracy. Where in the revised manuscript this change can be found – page 1, section 1, and line 30.

Comments 3: To enhance readability, consider relocating Table 1 to the end of the section discussing the CVD method or right before the Bioconvergence Modification Technology. Alternatively, if you opt to keep it in its current position, please provide expansions for LSE, EPD, CVD, and MAO within the table itself and include horizontal lines beneath each method to minimize reading confusion. Furthermore, it would be beneficial to include the advantages and disadvantages of Bioconvergence modification technology within the table for a more comprehensive overview.

Response 3: We thank the reviewers for their meaningful suggestions. We have changed the location of Table 1 to improve the readability of the article. Table 1 is now located at the end of the discussion of CVD methods or before biofusion modification techniques, which will help to better integrate the table with the content of the article. Regarding your suggestion to include the advantages and disadvantages of biofusion surface modification techniques in the table, we think this is a useful suggestion to provide a more comprehensive overview. However, according to you, Table 1 is primarily for comparing the advantages and disadvantages of methods for preparing surfaces, not the advantages and disadvantages of specific coatings. We agree with you that bioconvergence modification technology are mostly coating modification methods and do not fall within the scope of Table 1, so in subsequent sections of the article, we provide a detailed description of the advantages and disadvantages of coatings with polymerization modification techniques. Where in the revised manuscript this change can be found – page 15-16, section 3.5, and line 481-482.

Comments 4: There are abbreviation and expansion-related issues existing in the manuscript. For instance, in section 3.5, Chemical Vapor Deposition is abbreviated as CVD multiple times within the same paragraph. Similarly, there are instances where abbreviations like TNT are used without prior expansion at their initial mention. I recommend a thorough review of the manuscript to ensure that all abbreviations are adequately expanded upon their first use.

Response 4: We thank the reviewer for the meaningful suggestion. In response to your comments, we have thoroughly reviewed the article to ensure that all abbreviations are expanded on first use.

Comments 5: I suggest moving Table 3 right before the Future direction and challenges.

Response 5: We thank the reviewer for the meaningful suggestion. In response to your comments, we have moved Table 3 before the "Future direction and challenges" section to improve the structure and readability of the article. Where in the revised manuscript this change can be found – page 21-22, section 5, and line 690.

Round 2

Reviewer 1 Report

Comments and Suggestions for Authors

Good job!